# Epigenetic Changes in Gestational Diabetes Mellitus

**DOI:** 10.3390/ijms22147649

**Published:** 2021-07-17

**Authors:** Dominik Franciszek Dłuski, Ewa Wolińska, Maciej Skrzypczak

**Affiliations:** 1Chair and Department of Obstetrics and Perinatology, Medical University of Lublin, Street: Jaczewskiego 8, 20-954 Lublin, Poland; 2Department of Pathology, Medical University of Warsaw, Street: Żwirki i Wigury 61, 02-091 Warsaw, Poland; ewa.wolinska@wum.edu.pl; 3II Chair and Department of Gynecology, Medical University of Lublin, Street: Jaczewskiego 8, 20-954 Lublin, Poland; skrzypczakmk8@gmail.com

**Keywords:** gestational diabetes mellitus, DNA methylation, epigenetics, histone modification, non-coding RNAs

## Abstract

Gestational diabetes mellitus (GDM) is defined as carbohydrate intolerance that appears or is for the first time diagnosed during pregnancy. It can lead to many complications in the mother and in the offspring, so diagnostics and management of GDM are important to avoid adverse pregnancy outcomes. Epigenetic studies revealed the different methylation status of genes in pregnancies with GDM compared to pregnancies without GDM. A growing body of evidence shows that the GDM can affect not only the course of the pregnancy, but also the development of the offspring, thus contributing to long-term effects and adverse health outcomes of the progeny. Epigenetic changes occur through histone modification, DNA methylation, and disrupted function of non-coding ribonucleic acid (ncRNA) including microRNAs (miRNAs). In this review, we focus on the recent knowledge about epigenetic changes in GDM. The analysis of this topic may help us to understand pathophysiological mechanisms in GDM and find a solution to prevent their consequences.

## 1. Introduction and Overview of Gestational Diabetes Mellitus

Gestational diabetes mellitus (GDM) is a metabolic disorder involved in hyperglycemia defined as carbohydrate intolerance of any degree with onset or first recognition during pregnancy. This term was used for the first time by O’Sullivan in 1961. Its prevalence in a population is up to one in seven pregnancies worldwide [1]. GDM is associated with pregnancy and birth complications such as hypertension, preeclampsia, prematurity, fetal macrosomia, shoulder dystocia, and birth trauma [2,3]. There is a higher risk of prenatal and perinatal mortality, and the prevalence of c-section [3]. In addition, there are newborn complications such as hypoglycemia, hypoxia, or respiratory distress syndrome (RDS) [4]. The negative impact of GDM can be observed even in the later life of the mother and the child. Women with GDM have an increased risk of developing metabolic syndrome and type 2 diabetes mellitus (T2DM) [5,6]. The researchers estimate that up to 70% of patients with GDM will develop diabetes within 22–28 years after gestation [7,8,9,10]. The prevalence of GDM increases with risk factors such as obesity, increasing reproductive age of women, sedentary lifestyle (low physical activity, inadequate diet), and emerging environmental factors [7,8,9,10,11,12,13]. Similar problems, cardiovascular disease (CVD), metabolic diseases, obesity, and T2DM, were reported in offspring born to mothers with GDM during their adulthood [3,14,15]. There is also growing evidence linking GDM with abnormal brain development, with consequences such as general cognition [16] and concentration problems [17].

The International Association of Diabetes and Pregnancy Study Groups (IADPSG) established one of the most often used diagnostic criteria of GDM (Table 1). According to these recommendations, an oral glucose tolerance test (OGTT) with 75 g glucose is performed at 24–28 weeks of gestation. Three plasma glucose values are assessed: fasting, 1 h, and 2 h post glucose [18]. The existence of a strong association between maternal hyperglycemia and adverse pregnancy outcomes, which was analyzed in longitudinal studies, helps to individualize the cut-off values for the GDM diagnosis [3]. Although screening before 24–28 weeks of gestation is not recommended, a GDM diagnosis made as soon as possible is crucial to avoid pregnancy complications. [11,19,20]. The identification of phenotypes linked with poor outcomes, described as the presence of risk factors, may be presented as potentially useful in modulating and customizing the diagnostic and therapeutic aim in patients with GDM [21,22].

The main pathophysiological features of GDM are β-cell dysfunction and insulin resistance. Insulin sensitivity changes significantly during pregnancy, continuously adapting to the energy demands of both the mother and the fetus. It should be remembered that insulin sensitivity follows a biphasic course in healthy pregnant women; it starts from a sharp increase and then shows a marked decrease as the pregnancy progresses. The aim of metabolic adaptation observed during the first two trimesters is the storage of essential sources of energy, such as fatty acids and glucose necessary for the third trimester of pregnancy [23]. Estrogens, progesterone, human placental lactogen, and human placental growth hormone contribute to the progressive decline in insulin sensitivity. This insulin-resistance state in physiological gestation is an adaptive response, which favors the boost in free fatty acid and glucose blood levels, and it shifts the energy source from the mother to the fetus [24,25]. The β-cells of the pancreas mostly compensate by increasing the release of insulin. Hyperplasia and hypertrophy of β-cells, explained by increased proliferation and reduced apoptosis, were reported in human pregnancy by Butler et al. [26]. If dysfunction of β-cells appears, this compensatory effect disappears and GDM manifests itself. It was observed that the β-cell function is reduced by 30–70% in GDM [27].

Insulin resistance is a result of altered peripheral insulin signaling; glucose uptake is almost half that and insulin resistance is increased in GDM pregnancy compared to a physiological pregnancy [28]. Insulin signaling is affected by the impaired phosphorylation of the insulin receptor substrate 1 (IRS-1) or insulin receptor, although the number of receptors on the cell surface is the same [29]. Additionally, the development of insulin resistance is contributed to by pro-inflammatory cytokines such as tumor necrosis factor α (TNF-α), interleukin 1 (IL-1), and interleukin 6 (IL-6), which affect insulin signaling by inhibition of IRS-1 through serine phosphorylation [30,31,32,33]. The mechanisms that underline β-cell dysfunction are not fully discovered, but are probably similar to those in T2DM. Mostly, impairments at every step of insulin synthesis or secretion have been described; dysfunction of β-cells is activated by hyperglycemia and hyperlipidemia [34]. Mitochondrial dysfunction, oxidative stress, and endoplasmic reticulum stress are well-known consequences of gluco- and lipotoxicity, they affect insulin synthesis, secretion, and β-cell survival [35]. Furthermore, pro-inflammatory cytokines can induce β-cell de-differentiation and increase endoplasmic reticulum stress [36,37].

This review focuses on the latest knowledge on epigenetic changes in GDM. The analysis of this topic may help us to understand these complex pathophysiological mechanisms in GDM and find a solution as to how to prevent their consequences.

## 2. Programming and Epigenetic Mechanisms

Programming is the process whereby a stimulation at a critical window of development has long-term effects. The most important and well-known study is that of children born during the famine in the Netherlands in 1944–1945 [38].

Epigenetics refers to the processes of controlling gene activity that are not related to alterations of the deoxyribonucleic acid (DNA) sequence. The word epigenetics is of Greek origin and literally means over and above (epi) the genome. Generally, terms can be thought of as accents on words, where DNA is the language and modifications are the accent marks. Epigenetic marks change the way genes are expressed. The advantage of epigenetics is that it defines the cell differently than just by looking at gene expression levels. At the epigenetic level, each cell type will have specialized epigenetic patterns. The classical epigenetic mechanisms include histone modification and DNA methylation, but most authors also classify non-coding ribonucleic acids (ncRNAs) including microRNAs (miRNAs) as epigenetic modulators affecting the protein levels of target genes without altering their DNA sequence [39].

DNA methylation is the most extensively analyzed and studied epigenetic mechanism. During this process, a methyl group is added to the 5 position of cytosine residues. The enzymes that catalyze DNA methylation are called DNA methyltransferases (DNMT1, DNMT3A, and DNMT3B). DNMT1 is mainly responsible for maintenance of methylation during DNA synthesis. DNMT3A and DNMT3B are needed during de novo methylation [40]. These methylated cytosines are found first and foremost at the cytosine–phosphate–guanine (CpG), but can also be observed at non-CpG sites; both of them are important epigenetic indicators of genomic stability and gene expression [39]. In most cases, hypermethylation of DNA located in a gene promoter is connected with gene silencing. On the other hand, a low-level methylation is associated with gene activation. This type of change is essential for cell differentiation and mammalian development. Before the forming of blastocysts, the state of DNA methylation dramatically changes from hypermethylation to about 0% methylation. Next, methylation of DNA rises in a tissue-specific manner throughout pregnancy [41]. As development progresses, the rise in DNA methylation is preferential, and it occurs close to genes, that are involved in general development processes, while a decrease in DNA methylation is mapped to genes responsible for tissue-specific functions. Nevertheless, dynamic DNA methylation is linked to all regions of the gene, and this type of modification can regulate the gene expression in many ways during and after development. This type of modification can be negatively and positively associated with the gene’s expression. Furthermore, intragenic DNA methylation has been involved in assisting with the choice of exon–intron boundaries during co-transcriptional splicing of pre-mRNAs. On the other hand, there are suggestions in the literature that many changes in DNA methylation during development may not correlate with alternations in expression of the associated gene [42].

While most studies on epigenetics have focused on DNA methylation, we cannot forget about histone modifications. Histones may undergo several types of modifications, such as acetylation, methylation, or phosphorylation, that affect chromatin structure and gene expression. According to the current literature, histone modifications are crucial structural changes that suppress or promote gene expression, depending on the location of lysine or arginine residues in the histone. The acetylation of histones in most cases is associated with the active euchromatin state [43]. On the other hand, the correlation of methylation on histone 3 (H3) with transcription depends on its level and the residue, e.g., trimethylations of H3K9 and H3K4 are associated with the repression and activation of transcription, respectively. The transcription is affected otherwise by different levels of methylation on the same residue [44].

miRNAs are a class of small ncRNAs that regulate gene expression at the post-transcriptional level, affecting several target messenger RNAs (mRNAs) by binding to the 3 untranslated region (3UTR) of the mRNA within the miRNA seed region [45]. miRNAs and their target genes can regulate each other. Moreover, a single miRNA can downregulate the synthesis of hundreds of proteins [46].

## 3. Epigenetics in GDM

### 3.1. Epigenetics Linked with GDM—Key Observations

The evidence that epigenetics plays a crucial role in GDM is based upon the following observations:
GDM increases the risk of birth abnormalities in the offspring, but not all exposed fetuses are affected; some of them may develop in a proper way in spite of exposure [47,48,49]. This type of phenomenon, known as “partial penetrance”, suggests that even when the fetuses are genetically identical, non-genetic factors determine if a given offspring will have a birth defect [50].The severity of abnormalities can vary among the affected offspring, further highlighting non-genetic differences between fetuses with the same genetic background and the same exposure.Maternal diet is an important non-genetic risk factor for abnormalities in GDM [51,52], emphasizing the role of nutrition as a modulator of pathogenic processes during development [53].Transcriptional profiles are impaired in mouse embryos exposed to GDM [48], indicating a modified regulation of gene expression as a classic epigenetic mechanism.Changes between transcriptomic profiles are greater between exposed fetuses compared to non-exposed controls [54], indicating that gene-regulatory mechanisms are altered upon exposure.GDM and maternal obesity affect chromatin modifications, which provides potential substrates in impaired gene regulation [55].

Note, however, that additional studies are needed to show how the environment directly impacts the embryonic epigenome.

The influence of GDM on various aspects of pregnancy and offspring has been investigated in a number of experimental, preclinical and clinical models. The most important conclusions from the available literature data are described below and summarized in Table 2.

### 3.2. Pregnant Women with GDM Have a Different DNA Methylation Profile than in Non-GDM Pregnancies

Several studies confirmed a different DNA methylation pattern in GDM pregnancies when compared to healthy pregnant women. The DNA methylation in 42 fetal cord blood and 36 placenta samples was analyzed in a study by Awamleh et al. [75]. They identified 662 and 99 CpG sites in GDM placenta and cord blood, respectively. Among them, two sites for *AHRR* and *PTPRN2* were common for both sample types. The authors concluded that DNA methylation changes may represent adaptive placental and fetal responses to glucose intolerance. In another study, Deng et al. investigated methylation and gene expression changes in the visceral omental adipose tissue (VOAT) of pregnant women with GDM and controls using Illumina Human Methylation 450 k DNA Analysis Beadchip and whole Human Gene Expression Array. In the GDM group, 935 genes were dysregulated, among which 485 were upregulated and 450 DEGs were downregulated. Seven genes, *C10orf10*, *FSTL1*, *GSTT1*, *HLA-DPB1*, *HLA-DRB5*, *HSPA6*, and *MSLN*, were extracted. *C10orf10*, *FSTL1*, *GSTT1*, *HLA-DPB1*, and *HLA-DRB5* were hypermethylated with upregulated expression, while *HSPA6* was hypomethylated with downregulated expression. Only in the case of the *MSLN* gene, a typical negative correlation between gene expression and DNA methylation level was found, as significant hypermethylation in the CpG island and downregulated transcription were noted [56]. Similar differences were found in placenta samples analyzed in 32 GDM pregnant women and 31 controls [57]. The authors attempted to assess biological processes and mechanisms, which affect the occurrence and development of GDM, by the analysis of DNA methylation data and the gene expression profiles. As a result, 24,577 differential CpG sites mapping to 9339 DMGs and 931 DEGs between controls and GDM patients were found. Among these, *Oas1*, *Polr2g*, and *Ppie* were identified as possible pathogenic target genes of GDM [57].

In a work of Steyn et al., genome-wide RNA sequencing in maternal blood and placental tissue in GDM patients and non-GDM pregnancies was performed. Promoter region DNA methylation was examined for chosen genes and correlated with gene expression to examine epigenetic modifications. Reduced mRNA expression and increased DNA methylation were found for *G6PD* in GDM patients, and for genes encoding insulin-like growth factor (IGF)-binding proteins in GDM placentas [59].

Additionally, the methylation profile of genes that are involved in the inflammation process was investigated in peripheral blood samples from pregnant women with GDM. Halvatsiotis et al. found that only the *ATF2* gene was hypermethylated in GDM patients in comparison to healthy controls and preeclampsia (PE) patients. The genes encoding the interleukins and interleukin receptors *IL6R*, *IL4R*, *IL17RA*, *IL13*, *IL12A,* and *IL10RA* were significantly hypomethylated only in pregnant women with GDM [58].

To deeply investigate epigenetic changes that cause abnormal function of placental genes, Chen et al. performed in their study the multi-omic weighted gene co-expression network analysis (WGCNA) to identify the hub genes for GDM. They used epigenome- and transcriptome-wide microarray data, that were retrieved from the Gene Expression Omnibus (GEO) database. The results of WGCNA identified 15 modules and an MEblack module that had a statistically significant negative correlation with GDM. Gene Ontology (GO) enrichment analysis by the Biological Network Gene Ontology tool (BiNGO) of the MEblack module showed that these genes were primarily enriched for the regulation of interferon-α production, the interferon-γ-mediated signaling pathway, and presentation of antigen processing. Five hypermethylated, low-expression genes (*ABLIM1*, *GRHL1*, *HLA-F*, *NDRG1*, and *SASH1*) and one hypomethylated, highly expressed gene (*EIF3F*) were identified as GDM-related hub differentially methylated genes. This study revealed that placental function might be altered by dysmethylated hub genes, which influence GDM pathogenesis and fetal cardiac development [60].

### 3.3. The Exposure to GDM Induces Epigenetic Changes in Animal Offspring Models

In a study of Nazari et al., the impact of intrauterine exposure to GDM in the pancreatic islets on methylation of DNA, mRNA transcription, and protein expression was analyzed in Wistar rat offspring. The results show that hypomethylation of CpG sites in the neighborhood of cyclin-dependent kinase inhibitor 2A (CDKN2A) and cyclin-dependent kinase inhibitor 2B (CDKN2B) genes was positively correlated with higher levels of CDKN2A/B mRNA and protein in Langerhans islets in the 15-week-old GDM offspring. In the GDM group, the significantly lower average percentage of CDKN2A promoter methylation was observed in comparison to the control group, which may be linked with β-cell dysfunction and diabetes in the offspring [62]. Additionally, in a GDM mouse model, the changes in genomic DNA methylation and metabolic phenotypes in the pancreas of the offspring were investigated. It was found that in progeny of GDM mice, intrauterine hyperglycemia evoked glucose intolerance, insulin resistance, and dyslipidemia. The modified DNA methylation benchmarks in the pancreas and DMRs-related genes have been implicated in glycolipid metabolism and related signaling pathways, e.g., *Agap2*, *Cdh13*, *Fbp2*, *Gnas*, *Hnf1b*, *Irx3*, *Kcnq1*, *Lhcgr*, *Plcbr*, and *Wnt2*. The overall hypermethylation of *Agap2* was negatively correlated with the expression level of *Agap2* mRNA (*p* ≤ 0.001). A total of 338 differentially methylated regions (DMRs) in 20 chromosomes and differentially methylated genes (gene bodies and promoters) were identified. Eleven differentially methylated genes were downregulated and 12 were upregulated in the promoter region, and 97 were downregulated and 79 were upregulated in the gene body region. The results suggest that epigenetic modifications such as DNA methylation changes in the pancreas of GDM offspring might be involved in obesity, glycolipid metabolism abnormalities, and T2DM susceptibility in the adult GDM progeny [61].

### 3.4. Epigenetic Changes in GDM-Exposed Fetoplacental Endothelial Cells Can Affect Their Properties

In a pilot epigenetic study by Cvitic et al., an analysis of concordant DNA methylation and gene expression changes in GDM-exposed fetoplacental endothelial cells was conducted. Methylation and transcriptome analyses identified variants in gene expression involved in GDM-associated DNA methylation in 408 genes in arterial endothelial cells (AEC) and 159 genes in venous endothelial cells (VEC). The pathway analysis by Ingenuity Pathway Analysis (IPA) found genes which were impaired by exposure to GDM focused on functions associated with “cellular movement” and “cell morphology” in healthy VEC and AEC. The exposure to GDM programs atypical barrier function and morphology in fetoplacental endothelial cells by methylation of DNA and gene expression change. The effects were different in AEC and VEC and indicated a stringent cell-specific sensitivity to affected exposure-linked developmental programming in utero [63].

### 3.5. GDM Affects the Methylation Signature in the Cord Blood DNA and may Increase the Risk of Metabolic Disease in the Offspring

There are several studies showing an altered umbilical blood methylation profile in pregnancies in women with GDM. In a randomized clinical trial, data collected from 557 women recruited in 2009–2014 in the United Kingdom (UK) were investigated. Fasting glucose level, 1 h, and 2 h glucose concentrations following OGTT and GDM were found to be associated with 1, 592, 17, and 242 differentially methylated CpG sites, respectively, in the infant’s cord blood DNA. Cg03566881, a site located within the leucine-rich repeat-containing G-protein-coupled receptor 6 (LGR6), had the most significant correlation with GDM. Furthermore, the 1 h glucose level and GDM-associated methylation signatures in the cord blood of offspring appeared to be weakened by the physical activity and dietary intervention during pregnancy [66]. Similar findings were reported in a study of Haertle et al. that compared genome-wide methylation patterns of fetal cord blood from healthy pregnant women and GDM patients. Significant changes in methylation profiles between GDM and control blood samples were found at 65 CpG sites. Five genes, *ATP5A1*, *HIF3A*, *MFAP4*, *PRKCH*, and *SLC17A4*, were examined by bisulfite pyrosequencing. The effects were still significant even after modification for confounding factors such as gestational week, fetal sex, and maternal body mass index (BMI) in a multivariate regression model. Generally, the effect of fetal cord blood methylation was more evident in insulin-dependent GDM pregnant women than in women with dietetically treated GDM. These observed changes were of small effect size but affected multiple loci/genes. These genes could become useful markers for the diagnosis and management of adverse prenatal exposure [70].

In another genome-wide comparative methylome analysis using umbilical cord blood, Weng et al. conducted an analysis on samples from 30 offspring of GDM mothers and from 33 healthy mothers. Additionally, this quantitative analysis was verified by the validation of umbilical cord samples from 102 GDM infants and 103 control infants. A total of 4485 differentially methylated sites (DMSs), including 2335 hypomethylated and 2150 hypermethylated sites, were identified; 37 CpGs (from 20 genes) with a β-value difference of >0.15 between these two groups were found and presented their potential as clinical markers for GDM. The “hsa0490: type 1 diabetes” was the most important Kyoto Encyclopedia of Genes and Genomes (KEGG) pathway with *p*-values of 1.36 × 10^−2^ and 3.20× 10^−7^ in the hypomethylated and hypermethylated gene pathway enrichment analyses, respectively. Moreover, immune major histocompatibility complex (MHC)-related and neuron development-related pathways were significantly enriched in the GO pathway analyses. These results suggest that DNA methylation in GDM patients has an effect on genes that are preferentially associated with the type 1 diabetes mellitus (T1DM) pathway, neuronal development-related pathways, and MHC-related pathways, with implications for fetal development and growth. These data also provide supportive evidence that DNA methylation affects fetal programming [71].

In a smaller study of maternal blood and umbilical cord blood from 16 pregnant women, eight of whom had GDM, and their infants, Kang et al. showed significant differences in the methylation profile. They identified 381,869 variable methylation positions with statistically significant genome-wide differences in maternal blood, and 540,036 in cord blood. They selected 200 loci (167 corresponding genes in cord blood and 151 genes in maternal blood), which were differently methylated in the GDM and control groups. Additionally, several genes or molecules were identified that correlated with metabolic disease. Differences were significant for interleukin-6 (IL-6) and interleukin-10 (IL-10) in both groups; it was shown that single-nucleotide polymorphisms (SNPs) in the promoter of IL-10 correlated significantly with GDM development. Moreover, genes such as *ATP5A1, PRKCH,* or *MFAP4* had epigenetic signatures in cord blood that were significant between the GDM and non-GDM groups. In the authors’ opinion, the high-throughput platform could help analyze DNA methylation throughout the genome and identify the most propitious pathways and genes associated with GDM [74].

The intrauterine environment in GDM pregnancies may have a long-term effect in the offspring (Figure 1). To test the association of the epigenetic methylation profile with childhood adiposity-related outcomes, Yang et al. performed an epigenome-wide association analysis on whole blood samples from 162 offspring (81 GDM-exposed and 81 unexposed) enrolled in the EPOCH (Exploring Perinatal Outcomes in Children) study. The researchers also used 95 cord blood samples (31 GDM-exposed and 64 unexposed) from offspring participating in their own “Healthy Start” cohort. They identified 98 differentially methylated positions correlated with GDM exposure at a false discovery rate of <10% in peripheral blood. Fifty-one of these loci remained significant after correction of cell proportions. They also identified 2195 differentially methylated regions at a false discovery rate of <5% after correction of cell proportion. Then, they prioritized loci for pyrosequencing validation and association analysis with regard to adiposity outcomes based on previous publication, network and pathway analysis, strength of association and effect size, and analysis of cord blood. Methylation in 67% (in six out of nine) GDM-associated genes was validated and the methylation on SH3PXD2A was significantly associated with various adiposity-related outcomes [68].

The association between GDM and offspring complications such as diabetes or obesity was also investigated by Pinney et al. They demonstrated that changes in DNA methylation and gene expression in amniocytes that were exposed to GDM could be a mechanism leading to metabolic dysfunction later in life. They analyzed the expression of interferon-stimulated genes, which was higher in GDM amniocytes, describing 6 of the top 10 impaired genes. These biological pathways in GDM amniocytes included: the interferon response, inflammation, fatty liver disease, atherosclerosis and monogenic diabetes. Twenty DMRs were identified in female and forty-two in male GDM-exposed amniocytes. High-throughput chromosomes confirmation capture (Hi-C) analysis identified correlations between DMRs and 9 genes with significant expression alternations in female amniocytes and 11 in male amniocytes [65].

In a study by Wang et al., the effect of GDM on the expression and methylation of peroxisome proliferator-activated receptor γ-coactivator-1α (PGC-1α) and pancreatic and duodenal homeobox 1 (PDX1) in placenta and their effects on glucose metabolism in fetus were investigated. The analysis included 20 full-term placentas without complications of pregnancy and umbilical cord abnormalities and 20 cases of GDM placentas. Additionally, the levels of placental insulin, blood glucose, and glycosylated hemoglobin (HbA1c) were analyzed. The levels of PDX1 and PGC-1α mRNA were lower in the GDM group. The methylation level of the PGC-1α gene was significantly increased in the GDM group in comparison to controls. Glucose blood level was negatively correlated with the expression of PDX1 and PGC-1α mRNA in the placenta (r = −0.42, *p* = 0.01, r = −0,49, *p* < 0.01). Moreover, the placenta weight and the fetus birth weight in GDM patients were significantly increased compared to those in the control group. Blood glucose, insulin, and HbA1c levels were significantly increased in the GDM group. Blood glucose and HbA1c levels were correlated positively with insulin concentration, while insulin content was positively correlated with the placenta weight and the newborn birth weight. In their opinion, changes in the epigenetic modification of the *PGC-1α* gene in pregnant women with GDM may be a mechanism of abnormal glucose metabolism in the offspring [69].

To assess the effect of epigenetic mechanisms on the increased risk of development of metabolic disease in progeny mothers with GDM, a study on 1234 offspring (608 GDM and 638 controls) between 9 and 16 years from the Danish National Birth Cohort was conducted. The DNA methylation profile in the peripheral blood of 93 GDM and 95 control offspring was assessed. Additionally, the pyrosequencing for the validation of CpGs in 905 offspring was performed. A total of 76 differently methylated CpGs were identified in the GDM group (*p* < 0.05). Accounting for the BMI of the offspring did not change the relationship between GDM status and methylation levels for any of the 76 CpGs. Most of these DNA methylation modifications were associated with the maternal BMI before pregnancy, but 13 of them were independently correlated with GDM in mothers. The identified epigenetic changes might reflect the developmental programming of mechanisms of organ disease and may also be used as disease biomarkers [70].

There is evidence that the serotonergic system may be involved in obesity and metabolic disorders [76]. Blazevic et al., in their study, tested the hypothesis that GDM impairs the DNA methylation pattern of the fetal serotonin transporter gene (*SLC6A4*) and examined the functional importance of DNA methylation in regulation of *SLC6A4* expression in the human placenta. The study included 50 infants, 18 mothers with GDM, and 32 healthy mothers. All newborn were born by planned C-section. RNA and DNA were isolated from the fetal side of the placenta immediately after delivery. Polymerase chain reaction (PCR) amplification of bisulfate-treated DNA and subsequent DNA sequencing quantified DNA methylation at seven CpG sites in the *SLC6A4* distal promoter region. Reverse transcription–quantitative PCR (RT–qPCR) was used to measure *SLC6A4* mRNA levels. In the GDM group, the average DNA methylation of these seven loci was significantly decreased (*p* = 0.037) and showed a negative correlation with maternal glucose levels between 24 and 28 weeks of gestation (*p* < 0.05). Additionally, the average DNA methylation was inversely correlated with SLC6A4 mRNA levels in the placental tissue (*p* = 0.01). The results suggest that maternal metabolic status has an influence on DNA methylation of the fetal *SLC6A4* [73].

Cheng et al. investigated if the placental maternally expressed 3 (MEG3) DNA methylation profile correlated with maternal GDM status and offspring birth weight. MEG3 is an imprinted gene that encodes a long non-coding RNA and its epigenetic modification is related to diabetes. The study was conducted on blood samples from 46 pregnant women (23 with GDM and 23 controls). The results show that methylation levels in the MEG3 differentially methylated region (MEG3-DMR) differed significantly between GDM and controls on the maternal side of the placenta. Moreover, infant birth weight and maternal fasting glucose concentrations had a positive correlation with the mean MEG3 DNA methylation levels [67].

The aberrant DNA epigenetic modification may also affect the cardiometabolic risk increase in the offspring of mothers with GDM. In the work of Shiau et al., the influence of prenatal exposure to GDM on the offspring DNA methylation age in early childhood (at 3–10 years of age) was investigated. The results show that offspring exposed to GDM exhibit accelerated epigenetic age compared to healthy participants, which was also associated with cardiometabolic risk factors [64].

## 4. Conclusions

The presented data clearly show that epigenetic processes are not rare in gestational diabetes mellitus. Moreover, the obtained study results clearly indicate their influence not only on the course of pregnancy, but also on offspring and their further life. In addition, changes in the methylation profile are not limited only to metabolic pathways related to, among others, pancreatic β-cell dysfunction, the development of diabetes, dyslipidemia, or obesity, but also apply to other processes, such as inflammatory processes, neuronal development, MHC development, or the functioning of cellular pathways. A thorough understanding of the epigenetic disorders accompanying gestational diabetes, due to their often reversible nature, may allow us to avoid their undesirable consequences.

## Figures and Tables

**Figure 1 ijms-22-07649-f001:**
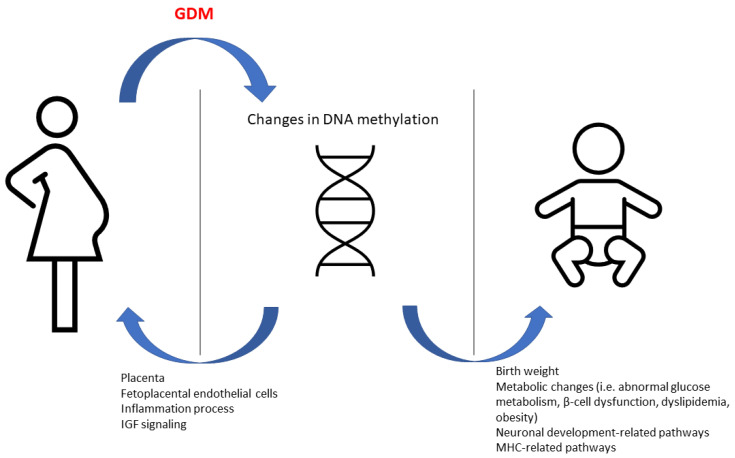
The effect of changes in DNA methylation in GDM women and their offspring.

**Table 1 ijms-22-07649-t001:** GDM diagnostic criteria according to the International Association of Diabetes and Pregnancy Study Groups (IADPSG) [18].

IADPSG Diagnostic Criteria of GDM (Any 1 of)	
fasting glucose, mmol/L (mg/dL)	≥5.1 (92)
1 h glucose, mmol/L (mg/dL)	≥10 (180)
2 h glucose, mmol/L (mg/dL)	≥8.5 (153)

**Table 2 ijms-22-07649-t002:** Epigenetic changes in various experimental GDM models described in the literature.

Study	Year of Publication	Experimental Model	Key Findings
Different DNA methylation profile in GDM vs. non-GDM pregnancies			
Deng et al. [56]	2018	Human	Antigen processing and presentation pathway and immune-related genes were associated with GDM in the visceral omental adipose tissue of pregnant women.
Zhang et al. [57]	2018	Human	*Oas1*, *Ppie*, and *Polr2g* as possible pathogenic target genes of GDM.
Halvatsiotis et al. [58]	2019	Human	*ATF2* gene was hypermethylated in GDM patients in comparison to healthy and PE patients. *IL6R*, *IL4R*, *IL17RA*, *IL13*, *IL12A*, and *IL10RA* genes were significantly hypomethylated only in pregnant women with GDM.
Steyn et al. [59]	2019	Human	Reduced mRNA expression and increased DNA methylation were observed for glucose-6-phosphate dehydrogenase gene *G6PD* in GDM patients, and for genes encoding insulin-like growth factor (IGF)-binding proteins in GDM placentas.
Chen et al. [60]	2020	Human	Dysmethylated genes (MEblack module) have a significantly negative correlation with GDM.
Epigenetic changes in GDM animal offspring models			
Zhu et al. [61]	2019	Mice	Altered patterns of DNA methylation were demonstrated in the pancreas of the GDM offspring.
Nazari et al. [62]	2019	Rat	Hypomethylation of CpG sites in vicinity of CDKN2A and CDKN2B positively correlates with increased levels of CDKN2A/B mRNA and protein in islets of Langerhans in GDM offspring.
Epigenetic effects on fetoplacental endothelial cells			
Cvitic et al. [63]	2018	Human	Variation in gene expression linked to GDM-associated DNA methylation on 408 genes in arterial endothelial cells, AEC, and 159 in venous endothelial cells, VEC. The exposure to GDM programs atypical morphology and barrier function in fetoplacental endothelial cells by DNA methylation and gene expression change.
Epigenetic changes in the cord blood and the effect on the offspring development			
Shiau et al. [64]	2021	Human	Prenatal GDM exposure causes accelerated offspring DNA methylation age in early childhood
Pinney et al. [65]	2020	Human	The expression of interferon-stimulated genes is increased in GDM amniocytes, which affects inflammation and interferon-related pathways. Additionally, novel differently methylated regions were found with potential distal regulatory functions.
Antoun et al. [66]	2020	Human	Maternal dysglycemia is associated with 1851 differentially methylated dmCpG sites in the infant’s cord blood DNA. This effect appears to be modified by a lifestyle intervention in pregnancy.
Cheng et al. [67]	2020	Human	DNA methylation levels in *maternally expressed gene 3* (*MEG3*)—DMR were significantly different between the GDM group and control group and correlated with maternal glucose concentrations and newborn birth weight.
Yang et al. [68]	2018	Human	Gestational diabetes mellitus exposure-associated DNA methylation assessment revealed 98 differentially methylated positions associated with GDM.
Wang et al. [69]	2018	Human	The levels of PGC-1α and PDX1 mRNA were lower in the GDM group; the methylation level of *PGC-1α* gene was higher in the GDM group.
Hjort et al. [70]	2018	Human	The exposure to GDM was associated with DNA methylation variation in 9- to 16-year-old offspring. A total of 76 differentially methylated CpGs in GDM offspring were identified compared to controls.
Weng et al. [71]	2018	Human	GDM has epigenetic effects on fetal growth and development; 37 CpGs were identified and showed potential as clinical biomarkers for GDM.
Haertle et al. [72]	2017	Human	A total of 65 CpG sites displayed significant methylation differences between GDM and controls.
Blazevic et al. [73]	2017	Human	Average DNA methylation across the 7 analyzed loci was decreased in the GDM group in comparison to controls.
Kang et al. [74]	2017	Human	A total of 151 loci in maternal blood group and 167 in cord blood group with different methylation in the GDM group compared to unexposed group.

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
