# Peer review of "Epigenetic Changes in Gestational Diabetes Mellitus"

_ijms, 2021, doi:10.3390/ijms22147649_

Round 1

Reviewer 1 Report

The review article “Epigenetic changes in gestational diabetes mellitus” is dedicated to  the analysis of recent knowledge about epigenetic changes in gestational diabetes mellitus. The article is well written.

The study has a good design.

The article is logically divided into sections and subsections.

In the article there are no grammatical and stylistic errors.

There is a table and many figures of good quality presented in the article.

The references cited are relevant and adequate.

A large number of scientific literature sources were analyzed.

The work has a high degree of novelty.

In my opinion, this review paper can be recommended for publication after minor revision.

It is recommended to separate section “Epigenetics and GDM” into subsections to make the article easier for understanding.

It is recommended to include a list of abbreviations, used in the article.

It is recommended to remove the gap between links 59 and 60 in the section "References".

It is recommended to add articles of  2021 to the list of references.

Reviewer 2 Report

Firstly, thank you for the opportunity to review your manuscript. However, I don't feel that it should be published in it's current form.

The content for the review is good, but there is no distillation into sub-sections. Table 1 is indicative of this issue - a number of studies are collected together, but there is no cohesive way that the papers are grouped. Following the table, there are paragraphs devoted to studies without looking at the common elements that involve GDM and epigenetics.

I began going through the paper, and have included some of the alterations and suggestions below. However, I feel that more effort needs to be made to distill the salient points. Also, it would be good to have/use figures to highlight some of these points.

----

Abstract: -

Too vague - Defining the two main topics, but not giving a flavour of what is to be covered by the review. The lines beginning “In this review, we focus on…” are the most specific part of the abstract.

This term was used FOR THE first time by O’Sullivan in 1961.

GDM is associated with pregnancy and birth complications SUCH as hypertension, preeclampsia, prematurity, fetal macrosomia, shoulder dystocia, birth trauma

IN ADDITION, THERE ARE We cannot forget about newborn’s complications as hypoglycemia, hypoxia, or respiratory distress syndrome (RDS)

There is also growing evidence linking GDM with abnormal brain development with its consequences like general condition [16] and concentration problems [17].

It's unclear what general condition means in the context of this sentence.

Instead of:

"The β-cells of the pancreas mostly compensate by an increased insulin release."

Try something like:

"The β-cells of the pancreas mostly compensate by INCREASING THE RELEASE OF INSULIN."

MiRNA = miRNA (this does not change whether it's at the beginning or within a sentence)

Round 2

Reviewer 2 Report

Thank you for making the requested changes - I feel you have made the review more comprehensive and easier to read.